# Potential Benefits of Combining Proton or Carbon Ion Therapy with DNA Damage Repair Inhibitors

**DOI:** 10.3390/cells13121058

**Published:** 2024-06-19

**Authors:** Gro Elise Rødland, Mihaela Temelie, Adrian Eek Mariampillai, Sissel Hauge, Antoine Gilbert, François Chevalier, Diana I. Savu, Randi G. Syljuåsen

**Affiliations:** 1Department of Radiation Biology, Institute for Cancer Research, Norwegian Radium Hospital, Oslo University Hospital, 0379 Oslo, Norway; 2Department of Life and Environmental Physics, Horia Hulubei National Institute of Physics and Nuclear Engineering, 077125 Magurele, Romania; 3UMR6252 CIMAP, Team Applications in Radiobiology with Accelerated Ions, CEA-CNRS-ENSICAEN-Université de Caen Normandie, 14000 Caen, Francefrancois.chevalier@ganil.fr (F.C.)

**Keywords:** high-LET irradiation, DNA repair, DNA damage response inhibitors, antitumor immune response, radiosensitivity

## Abstract

The use of charged particle radiotherapy is currently increasing, but combination therapy with DNA repair inhibitors remains to be exploited in the clinic. The high-linear energy transfer (LET) radiation delivered by charged particles causes clustered DNA damage, which is particularly effective in destroying cancer cells. Whether the DNA damage response to this type of damage is different from that elicited in response to low-LET radiation, and if and how it can be targeted to increase treatment efficacy, is not fully understood. Although several preclinical studies have reported radiosensitizing effects when proton or carbon ion irradiation is combined with inhibitors of, e.g., PARP, ATR, ATM, or DNA-PKcs, further exploration is required to determine the most effective treatments. Here, we examine what is known about repair pathway choice in response to high- versus low-LET irradiation, and we discuss the effects of inhibitors of these pathways when combined with protons and carbon ions. Additionally, we explore the potential effects of DNA repair inhibitors on antitumor immune signaling upon proton and carbon ion irradiation. Due to the reduced effect on healthy tissue and better immune preservation, particle therapy may be particularly well suited for combination with DNA repair inhibitors.

## 1. Introduction

Radiotherapy is frequently used in cancer treatment. In traditional radiotherapy, high-energy X-rays are employed. However, particle radiotherapy with protons or carbon ions has become more widely available in recent years [1]. The main reasoning for introducing particle radiotherapy is the beneficial depth dose distribution of particles. Most of the particles’ energy is deposited at the end of their trajectory as they are slowed down, causing a sharp dose peak named the Bragg peak [2] (Figure 1A). By directing this peak to the tumor, particle irradiation can thus result in improved sparing of surrounding normal tissue, reducing treatment-related side effects [2,3]. The beam energy determines the depth of the Bragg peak, and by using beams with different energies in the so-called spread-out Bragg peak (SOBP), one can cover the whole tumor volume with a uniform dose (Figure 1B).

Another notable feature of particle radiotherapy is the elevated ionization density, leading to more clustered DNA damage. The ionization density reflects the absorbed radiation dose per unit length, also termed linear energy transfer (LET). As protons or carbon ions traverse tissue, they are slowed down, and the probability for ionizations increases. There is therefore a gradual increase in LET from the beam entrance towards the end of the Bragg peak. Clinical carbon ion beams produce high-LET radiation, with typical LET values within the spread-out Bragg peak (SOBP) of 40–80 keV/µm [4] and >200 keV/µm at the distal end [5,6], depending on the beam energy. For clinical proton beams, average LET values are approximately 2–3 keV/µm at the middle of the SOBP and 10–15 keV/µm at the distal end of the Bragg peak [7,8]. In comparison, the LET value for a clinical 6 MV X-ray beam is about 2 keV/µm [9]. The average LET values of proton beams are thus similar or modestly elevated compared to those of X-ray beams. However, these are average values, and proton beams exhibit heterogeneity due to the multiple proton energies present in the SOBP [10,11]. This can lead to a considerable amount of proton radiation with higher LET values, even at the middle of the SOBP [11].

The clustered DNA damage may consist of several DNA double-strand breaks and/or other types of DNA lesions in close proximity to each other (also termed complex DNA damage, see Figure 1B). This damage may be particularly hard to repair and often results in cancer cell death [12]. Importantly, the ionization density is highest at the very distal end of the Bragg peak, typically located in the surrounding margin outside the tumor volume. The clustered damage could thus in some cases affect nearby critical normal tissues, resulting in increased adverse effects. There is therefore a high need to better understand how cells respond to DNA damage caused by the high-LET particle irradiation, both in order to find ways to enhance tumor cell killing and to avoid radiation-induced side effects [1,10].

Previously, a somewhat common assumption has been that DNA repair is not very important following high-LET irradiation, as the clustered DNA damage has been considered almost non-reparable [13,14,15]. Upon high-LET irradiation, the shape of the survival curve is typically straight-lined at low doses, indicating lack of repair [15]. Furthermore, the cell cycle is decoupled from radiosensitivity as LET increases [16], further indicating that DNA repair is not contributing much to cell survival after high-LET irradiation. This view was, e.g., supported by previous experiments showing higher relative biological effectiveness (RBE) for repair-proficient than repair-deficient (Ku80 negative) Xrs5 Chinese hamster ovarian (CHO) cells after irradiation with carbon ions [17]. Consequently, it was proposed that the benefit of carbon ion therapy, over conventional X-ray therapy, would be largest in tumors with high repair capacity [17].

However, more recent studies indicate that DNA repair is highly important even after high-LET irradiation [18,19]. Combination treatment with DNA repair inhibitors is thus relevant for both carbon ions and protons. Here, we discuss potential differences in repair signaling after high- and low-LET irradiation that may impact such treatments (Figure 2). We also examine previous preclinical work involving DNA repair inhibitors combined with proton or carbon ion irradiation. Moreover, we discuss how such combination treatment may promote antitumor immune signaling. We propose that DNA repair inhibitors are useful in combination with particle radiotherapy, causing tumor radiosensitization and enhancing antitumor immune effects.

## 2. Differences in Repair Signaling for High- and Low-LET Irradiation

The question of whether high- and low-LET radiation induces different repair pathways has not yet been conclusively answered. Interestingly, many previous studies have shown a delayed resolution of DNA repair foci after high-LET irradiation, as assessed through immunofluorescence microscopy. For example, a higher number of persistent γH2AX and 53BP1 foci, two commonly used markers for DNA double-strand breaks, has been observed following irradiation with carbon ions compared to X-irradiation [20,21,22,23]. Additionally, a slower resolution of 53BP1 and γH2AX foci was found in cells irradiated at the distal end compared to the front or center position of the Bragg peak for clinical proton beams [24,25]. These studies suggest that DNA repair may be slower following high-LET compared to low-LET irradiation. Notably, slower repair could likely involve different use of repair pathways, meaning that specific pathways are involved in the repair of the clustered damage induced by high-LET particles.

On the other hand, the γH2AX and 53BP1 foci caused by high-LET irradiation are large and may not represent single DNA double-strand breaks [20,26]. Thus, repair may occur on a subset of the breaks without the foci disappearing, leading to the underestimation of repair rates. Interestingly, the assessment of DNA double-strand breaks by pulse-field gel electrophoresis has shown rapid rejoining for most of the breaks even after irradiation with helium and nitrogen ions with very high LET values ranging from 80 to 225 keV/µm [27]. The measurement of DNA double-strand breaks via the neutral comet assay has also revealed similar repair rates for protons with different LETs (1 and 12 keV/μm) [28]. It is noteworthy that in the latter study, when applying the alkaline comet assay, slower repair was observed for the high-LET protons. This suggests that high-LET irradiation may cause a slower resolution of single-strand breaks/alkali-labile sites [28].

Consistent with the notion that specific repair pathways may be involved in the repair of DNA double-strand breaks after high-LET irradiation, it has been proposed that clustered DNA damage generally induces more DNA end resection [12]. This is a process where nucleases such as MRE11, EXO1, and CtIP (RBBP8) generate single-stranded DNA at sites of DNA double-strand breaks, thereby facilitating the binding of specific repair proteins [29]. Accordingly, DNA double-strand break repair pathways requiring end resection, such as alternative non-homologous end joining (alt-NHEJ) and homologous recombination (HR), may be more important after high-LET than low-LET irradiation [12]. The classical NHEJ (c-NHEJ) repair pathway, by contrast, does not require end resection [29]. In support of increased end resection after high-LET irradiation, more foci of the single-strand binding protein RPA were observed in G2 phase cells following carbon ion compared to X-irradiation [23]. Furthermore, exposure to heavy ions with a broad spectrum of LET values ranging from 90 to 15,000 keV/µm showed a clear positive correlation between the number of RPA foci with increasing LET in both G2 and G1 phase cells [30]. The increased resection in the G1 phase likely promotes error-prone alt-NHEJ [30]. It has also been reported that high-LET irradiation leads to a more pronounced activation of the ATR-dependent G2 checkpoint [31,32,33,34,35]. As activation of the ATR kinase may depend on end resection mechanisms [36], this further supports that end resection is increased.

Several studies have demonstrated a higher dependency on HR repair after high-LET as compared to low-LET irradiation, for instance in comparisons of carbon ion and X-irradiation [37,38]. Increased dependency on HR repair has also been reported for proton irradiation at relatively low average LET values of less than 10–15 keV/µm [39,40,41,42]. However, other studies did not find such correlations between HR repair and LET [43], or the effects varied between different cell lines [21]. The link between LET and HR repair is thus not entirely clear [44,45]. In addition, the role of c-NHEJ after high-LET irradiation is incompletely understood. Intriguingly, it has been shown that small DNA fragments associated with clustered DNA damage suppress the binding of KU70/80 (XRCC6/5), the initial step of the c-NHEJ pathway [46,47]. This suggests a reduced involvement of c-NHEJ in repairing damage caused by high-LET irradiation. Notably, electron microscopy studies have shown that particle tracks induce the largest DNA damage clusters in densely packed chromatin [26,48]. As large clusters may yield the highest number of small fragments, the inhibition of NHEJ by small DNA fragments may likely occur specifically in regions of heterochromatin [48]. Conversely, other studies have shown that c-NHEJ remains the major repair pathway following both low- and high-LET irradiation. It was found that the loss of c-NHEJ gave bigger radiosensitizing effects compared to the loss of HR, even for carbon ion and other heavy ion irradiation [38,49,50].

As mentioned above, the clustered damage sites may contain different types of DNA damage. Therefore, the repair of base damage or single-strand breaks may also be important in addition to the DNA double-strand break repair pathways. Indeed, a recent study showed that the base excision repair protein 8-oxoguanine DNA glycosylase (OGG1) contributes to radioresistance after high-LET (12 keV/µm) proton irradiation but not low-LET (1 keV/µm) proton or X-irradiation [51]. Furthermore, repair focus analysis, with antibodies to OGG1 as a surrogate marker for base damage, showed that these foci were more persistent after high-LET as compared to low-LET proton irradiation [51]. In cells irradiated with high-LET iron ions, persistent OGG1 foci co-localized with XRCC1 and 53BP1 foci, in agreement with the repair of clustered damage being dependent on multiple repair pathways [52]. An interesting issue is whether the clustered damage sites may cause obstacles to the replication fork, potentially creating additional DNA double-strand breaks when cells undergo DNA replication several hours after irradiation. It has been demonstrated that the loss of SLX4-MUS81, which is involved in resolving stalled forks, leads to a bigger sensitization of cells irradiated with protons than with X-rays [53]. The loss of FANCD2, which binds SLX4, also sensitizes cells to protons [11]. Although these factors may also be involved in regulating HR repair of the radiation-induced breaks, one intriguing explanation could be that proton-induced clustered DNA damage causes more stalling of the DNA replication forks, resulting in increased DNA breakage when cells are lacking SLX4-MUS81 or FANCD2 [53]. The loss of PARG, which also plays roles in resolving stalled replication forks, has been shown to cause bigger sensitization to protons than to X-rays [51]. Moreover, DNA fiber assays have revealed a bigger increase in replication stalling following proton compared to X-irradiation of *KRAS*-mutated cells [54]. However, the study did not find similar effects in *KRAS* wild-type cells. More work is needed to understand the interaction between DNA replication and clustered damage sites following high-LET irradiation.

## 3. Previous Studies of DNA Repair Inhibitors Combined with Protons or Carbon Ions

Several inhibitors involved in DNA repair are under clinical investigation as radiosensitizers in combination with traditional X-ray radiotherapy. However, their effects in combination with particle radiotherapy have not been fully elucidated. Below we discuss the radiosensitizing effects observed in recent preclinical studies with carbon ions or protons in combination with inhibitors of various DNA damage response proteins.

### 3.1. PARP Inhibitors

It is well known that the loss of function of HR repair sensitizes cells to PARP inhibitors [55]. In cells defective in HR, DNA repair is performed by other processes such as NHEJ, alt-NHEJ, and single-strand break repair [56]. Several of these alternative repair pathways are regulated by PARP, leaving the cells sensitive to PARP inhibition [56]. Indeed, when PARP is inhibited, it leads to single-strand break accumulation, stalling of the replication fork, and the generation of many toxic double-strand breaks [57]. These double-strand breaks are poorly repaired, if at all, with error-prone repair mechanisms. This leads to genetic instability and eventually cell death, by synthetic lethality [57,58]. Combining irradiation with PARP inhibitors offers a promising approach for cancer treatment [57,58,59,60]. Radioresistant cancer cells can often have alterations in the HR or other DNA repair pathways [61], leaving them sensitive to PARP inhibitors which can help overcome this resistance [62,63]. The effect is even more pronounced using high-LET irradiation, such as carbon ions and protons [64]. The inhibition of PARP prevents the repair of single-strand breaks, while high-LET radiation induces more double-strand breaks, altogether overwhelming the compromised repair machinery of cancer cells with defective HR. Additionally, PARP is required for alt-NHEJ [12], which is thought to be particularly important for the repair of DNA damage following high-LET irradiation (as mentioned in Section 2).

The combination of radiation with various LETs and PARP inhibitors has been tested in several cellular models. A study comparing olaparib treatment combined with Bragg peak protons (LET: 4.3 keV/μm) versus entrance beam protons (LET: 0.5 keV/μm) indicated that the most effective radiosensitization occurred with the higher LET of Bragg peak protons in lung and pancreatic cancer cells [65]. Similar results were obtained with HeLa cells in another study, where olaparib treatment was combined with protons with LET values of 12 keV/μm versus 1 keV/μm [66]. Even for BRCA-proficient breast cancer cells, greater radiosensitization was observed with protons (LET within 3.0–8.1 keV/μm) than with X-rays when combined with olaparib [67]. Moreover, in 3D spheroid models of oropharyngeal squamous cell carcinoma, both olaparib and talazoparib were found to sensitize to both low-LET protons (~1 keV/µm) and X-rays. Talazoparib exhibited enhanced efficacy over olaparib specifically for proton irradiation, possibly due to talazoparib’s stronger PARP trapping ability [68]. Olaparib also sensitized pancreatic ductal adenocarcinoma xenograft models to chemoradiotherapy in vivo, with greater radiosensitizing effects observed with SOBP protons (LET within 2–5 keV/μm) compared to X-rays [69]. The combination of olaparib and proton irradiation was also effective in platinum- and radiation-resistant esophageal cancer [70]. Niraparib, another PARP inhibitor, increased the RBE of protons versus photons in human head and neck squamous cell carcinoma (HNSCC) cell lines [71]. (In the two latter reports the proton LET was not mentioned).

A comparison of high-LET (>100 keV/μm) carbon ion irradiation and low-LET gamma radiation reported greater radiosensitization with PARP inhibition for the carbon ions [72]. Additionally, triple-negative breast cancer cell lines irradiated with carbon ions (80 keV/μm) or X-rays in combination with olaparib showed radiosensitization mainly for *BRCA*-mutated cells, and the sensitizing effect was bigger for carbon ions than for X-rays [73]. Carbon ions (50 keV/μm) in combination with talazoparib also reduced the stem cell proportion in radioresistant glioma [74]. A comparison of X-rays, protons (11 keV/µm), and carbon ions (73 keV/µm) in combination with olaparib showed a greater impact of high-LET carbon ion irradiation on chondrosarcoma cell survival, but radiosensitization with olaparib was observed for all types of radiation. The sensitizing effect was biggest for protons, followed by carbon ions and X-rays [75]. In summary, numerous previous studies suggest that the radiosensitizing effects of PARP inhibitors are more pronounced for carbon ion and proton irradiation than for X-irradiation.

### 3.2. ATR, CHK1, and WEE1 Inhibitors

The ATR kinase and its downstream target CHK1 kinase mediate G2 checkpoint arrest and HR repair after irradiation and are also the main responders to replication stress [76,77,78,79,80,81]. Similarly, the WEE1 kinase is a crucial regulator of the G2 checkpoint, HR repair, and replication stress [77,78,82,83,84]. If HR is indeed particularly important for the repair of clustered DNA damage, and such damage potently stalls replication forks, inhibitors of ATR, CHK1, or WEE1 may be particularly good radiosensitizers for high-LET radiation. The stronger activation of the G2 checkpoint after high-LET irradiation may further prove a greater efficacy of these inhibitors in combination with high-LET particles.

However, studies comparing X-irradiation with proton and carbon ion irradiation in combination with ATR inhibitors have most often reported smaller or equal radiosensitizing effects for the highest LET irradiation. For instance, treatment of cancer cells with the ATR inhibitor AZD6738, in combination with protons of two LET values (1 and 7 keV/μm) or X-rays, revealed equal radiosensitization for all radiation types [85]. Furthermore, although cancer cells treated with VE-821, another ATR inhibitor, were sensitized to carbon ion irradiation (LET: 70 keV/µm) [86], the sensitization enhancement ratio (SER) was slightly lower than that seen for X-rays. ATR inhibition also led to the abrogation of the G2 checkpoint and increased formation of micronuclei. The data in this study are in good agreement with our own data showing G2 checkpoint abrogation and decreased clonogenic survival upon the co-treatment of glioblastoma cells with the ATR inhibitor VE-822 and carbon ion irradiation of two LET values (28 and 73 keV/μm), but with a slightly smaller effect for the higher LET value as compared to the lower LET value and X-irradiation [33]. A study of chondrosarcoma cell proliferation has also reported a sensitizing effect of VE-821 combined with carbon ion irradiation (LET: 55 keV/µm) [87]. The sensitizing effect was higher than that observed for relatively low-LET protons (2.9 keV/μm), but lower than for X-rays. In HPV-negative HNSCC cell lines in 2D culture, the sensitizing effect of VE-821 was diminished even by low-LET proton irradiation (~1 keV/µm) as compared to that observed in X-irradiated samples [88]. However, the same study showed that growth of tumor spheroids was delayed by ATR inhibition after exposure to both radiation types. Another study comparing protons of a relatively high LET (9.9 keV/µm) with X-rays in combination with AZD6738, found that the RBE measured by clonogenic survival was increased in two out of four cancer cell lines [89]. 

Several studies have also addressed the effects of CHK1 inhibitors in combination with X-irradiation versus proton or carbon ion irradiation. In breast cancer cell lines irradiated at the SOBP of a 230 MeV proton beam, clonogenic survival seemed to be equally well reduced in the proton and X-irradiated cells upon co-treatment with the CHK1 inhibitor PF-477736 [90]. Similar results were observed in 3D pancreatic tumor models treated with a different inhibitor (prexasertib) in combination with low-LET protons (3.7 keV/µm) and X-rays [91]. Another study reported that clonogenic survival in one out of two lung cancer cell lines was decreased by co-treatment with the CHK1 inhibitor (AZD7762) and carbon ions (50 keV/µm), but not X-irradiation [92]. Furthermore, a synergistic effect between high-LET carbon ion (184 keV/µm) irradiation and the CHK1 inhibitor UCN-01 was observed in cancer stem-like cells of HNSCC [93]. Synergy was observed for comparatively more doses of carbon ion than of X-radiation. However, in both studies with carbon ions, the radiosensitizing effects of CHK1 inhibition were very small. Reports on WEE1 inhibition in combination with high-LET irradiation are sparse. However, one study showed that the WEE1 inhibitor AZD1775 caused the abrogation of G2 arrest and reduced survival in lung cancer cells irradiated with carbon ions (50 keV/μm) [94]. The radiosensitizing effect was, however, quite low, and lower than that observed for X-irradiated cells in combination with the WEE1 inhibitor. Taken together, based on the literature to date, the radiosensitizing effects of ATR, CHK1, and WEE1 inhibitors do not appear to be bigger for high-LET than for low-LET radiation.

### 3.3. DNA-PKcs Inhibitors

The DNA-dependent protein kinase (DNA-PK) catalytic subunit (DNA-PKcs) is the key driver of the c-NHEJ pathway. The DNA-PKcs is also implicated in mitosis, telomere maintenance, and other cellular processes [95,96]. As mentioned above, it has been suggested that c-NHEJ may be less involved in the repair of DNA damage caused by high-LET irradiation, due to the inhibiting effect of small DNA fragments (see Section 2). One might thus expect that DNA-PKcs inhibitors would not likely work well in combination with high-LET carbon ion irradiation. However, preclinical studies have reported that DNA-PKcs inhibitors can act as radiosensitizing agents to protons as well as carbon ions. The potent DNA-PKcs inhibitor NU7026 significantly enhanced H2AX phosphorylation and reduced clonogenic survival in non-small cell lung cancer H1299 cells exposed to carbon ion (50 keV/µm) and X-radiation [49], although with a slightly reduced SER for carbon ions. The same inhibitor increased the sensitivity of lung cancer A549 cells to carbon ion radiation (49 keV/µm), enhancing cell death, reducing DNA damage repair, exacerbating cell cycle G2/M phase arrest, and increasing apoptosis [97]. In comparison with X-rays, apoptosis seemed to be more pronounced with carbon ions. In a study by Bright et al. [89], the effects of NU7441, another DNA-PKcs inhibitor, in parallel with other inhibitors targeting the DNA damage response (e.g., ATM, ATR, RAD51, PARP), was investigated in combination with proton (9.9 keV/µm) or X-irradiation in lung cancer (H460 and H1299), pancreatic cancer (PANC-1 and Panc 10.05), and normal endothelial (HUVEC) cells. NU7441 was the only inhibitor that caused a significant radiosensitization of all cell lines, but with either higher or lower SER for protons than for X-rays depending on the cell line. In studies comparing treatment of 3D cancer cell cultures with low-LET protons (3.7 keV/μm) versus X-rays in combination with NU7026, similar radiosensitization was observed for protons and X-rays [91,98]. The radiosensitizing effect of the DNA-PKcs inhibitor M3814 to carbon ion irradiation has also been studied under hypoxic (1% O_2_) compared to normoxic (21% O_2_) conditions. While the oxygen enhancement ratio for the X-irradiation of A549 cells was 1.4, no significant oxygen effect was found after carbon ion irradiation (LET value not reported, but irradiation was done at SOBP). Interestingly, the radiosensitizing effect of M3814 was found to be stronger at hypoxia than normoxia when combined with carbon ion irradiation [99]. These reports highlight the ability of DNA-PKcs inhibitors to enhance cell death by particle irradiation, likely through the inhibition of c-NHEJ. However, the underlying mechanism is not completely clear. There is evidence that both NU7026 and NU7441 increase the radiosensitivity of cancer cells to carbon ion therapy independently of DNA DSB repair [100,101]. DNA-PKcs inhibition by NU7026 led to rapid telomere length shortening together with extensive senescence in MCF-7 breast cancer cells exposed to carbon ion radiation [100]. Furthermore, NU7441 radiosensitized H1299 lung cancer cells via a significant G2/M cell cycle arrest and a higher level of senescence, with no apparent inhibition of DNA repair [101]. Thus, more studies using protons and carbon ions are needed to uncover the detailed mechanisms of radiosensitization by DNA-PKcs inhibitors for therapeutic optimization.

### 3.4. ATM Inhibitors

The ATM kinase is involved in the repair of DNA double-strand breaks, being considered a key protein in the HR pathway [102]. ATM also promotes resection-dependent repair of DNA double-strand breaks during the G1 phase and serves as a major regulator of cell cycle checkpoints [103]. While many previous studies have demonstrated radiosensitization by ATM inhibitors in combination with X-rays in a variety of preclinical tumor models, only a few studies have been performed with a focus on the LET dependency of the effects.

In one study, the ATM inhibitor AZD0156 exhibited greater radiosensitization of cancer cells exposed to proton radiation at the Bragg peak (7 keV/μm) of a 76.8 MeV spot scanning proton beam than to X-rays or low-LET protons (2.2 keV/µm; irradiated at the proton beam entrance). This radiosensitization was found to depend on the NHEJ repair proteins XRCC4 and LIG4, suggesting that ATM inhibition in combination with Bragg peak irradiation may lead to the increased use of NHEJ with toxic consequences [85]. Interestingly, this study noted increased phosphorylation of CHK1 and RPA32, but not of CHK2, following Bragg peak proton irradiation as compared to X-irradiation, indicating heightened activation of ATR signaling rather than ATM signaling. However, only ATM inhibition selectively sensitized cells to Bragg peak protons. In the abovementioned study by Bright et al., with protons of LET 9.9 keV/μm in comparison to X-rays, it was observed that KU55933, another ATM inhibitor, sensitized H460 lung cancer cells more to protons than X-rays, but H1299 lung cancer cells were sensitized more to X-rays than protons [89]. The radiosensitizing effects seen in the aforementioned studies in 3D cancer cell cultures [91,98] were largely similar for treatment with KU55933 in combination with low-LET protons (3.7 keV/μm) and X-rays. Similarly, a study on colorectal cancer organoids with the two ATM inhibitors AZ32 and KU55933 showed roughly similar sensitization to both proton and X-irradiation. Although the LET of protons was not specified in this study, it was likely relatively low, as irradiation was conducted in the middle of the SOBP [104]. Additionally, a study treating uveal cancer cells with KU55933 in combination with X-rays and low-LET protons (1 keV/µm) also demonstrated comparable sensitization to both types of radiation [105]. Recently, we showed that AZD1390, another potent ATM inhibitor, could radiosensitize glioblastoma cells to both X-rays and carbon ions (28 and 72 keV/μm), albeit with slightly lower sensitization observed for the higher-LET carbon ions as compared to X-rays [33]. ATM inhibitors, with their low single-agent toxicity, may show clinical success as sensitizers for both X-rays and particles. It is worth mentioning, however, that the ATM inhibitor KU55933 significantly reduced the clonogenic survival of normal human fibroblasts irradiated with carbon ions (LET: 70 keV/µm) in vitro [106].

## 4. Antitumor Immune Signaling Induced by DNA Repair Inhibitors

Recent studies have shown that combination treatment of DNA repair inhibitors and X-rays can yield increased antitumor immune effects relative to those seen after irradiation alone. This can occur through several different mechanisms. A type 1 interferon (IFN) response may be elevated due to enhanced micronucleus formation through the activation of the cGAS/STING/TBK1/IRF3 pathway [107,108]. Micronuclei are formed when cells with unrepaired or misrepaired DNA damage undergo mitosis [80,109], and the abrogation of cell cycle checkpoints or defective repair can enhance micronucleus formation after irradiation [80,109,110,111]. Multiple studies have shown that ATR, PARP, and CHK1 inhibitors can increase the IFN response through this mechanism after irradiation [110,111,112,113,114,115]. Interestingly, after treatment with ATR or ATM inhibitors and X-rays, the cytosolic RNA sensor RIG-1 may also initiate IFN signaling [116,117]. Cytosolic RNA may arise from the activation of transposable elements [118,119], resulting from the opening of compacted chromatin and transcription of DNA that is normally silent [120]. Cytosolic RNA or DNA can also be released from mitochondria, as has been shown in cells irradiated with X-rays [121]. Inhibition of ATM in non-irradiated cells has also been shown to facilitate the release of mitochondrial DNA [122]. Another way of increasing immune responses is by inducing immunogenic cell death, in which dying cells present damage-associated molecular patterns (DAMPs) that stimulate dendritic cells and other immune cells [123]. Some studies report the increased presentation of immunogenic cell death markers HMGB1, ATP, and/or calreticulin upon treatment with X-rays and either ATR inhibitors in vitro [124] or PARP inhibitors in vitro and in vivo [113,125]. DNA repair inhibitors have also been shown to regulate immunosuppressive effects after X-irradiation. For example, ATR inhibition impairs the radiation-induced upregulation of the immune checkpoint molecules PD-L1 [126,127,128] and CD47 [126].

As of now, very few studies have addressed how combined treatment with DNA repair inhibitors and high-LET particle radiation affects immune responses, but the antitumor immune signaling could likely be enhanced (Figure 3). We recently observed higher levels of secreted IFN-β from cells treated with an ATR inhibitor in combination with carbon ions and high-LET protons (38 keV/µm) as compared to low-LET protons and X-rays [33]. The mechanism underlying the increased IFN response when the ATR inhibitor was combined with high-LET irradiation is not known. However, higher-LET radiation alone has been shown to induce higher levels of micronuclei as compared to lower-LET radiation [129,130]. Combined treatment with DNA repair inhibitors and proton or carbon ion radiation may therefore induce the formation of even more micronuclei than the combination of such inhibitors with X-rays. Indeed, one study observed increased radiosensitization and micronucleus formation when an inhibitor of ATR was combined with high-LET carbon ion (70 keV/µm) as compared to X-irradiation [86]. Increased micronucleus formation following DNA repair inhibition and high-LET irradiation could thus likely result in higher IFN-β levels. Additionally, in a recent study, higher IFN-β signaling after proton than photon irradiation was associated with proton-induced activation of transposable elements, likely due to the aforementioned opening of compacted chromatin [131]. As non-repaired clustered DNA damage can cause changes in chromatin structure [132], it is thus tempting to speculate that co-treatment with DNA repair inhibitors and high-LET radiation could result in the further opening of compacted chromatin, and thereby heighten the transcription of retrotransposons or other transposable elements. This could likely result in cytosolic dsRNA that stimulates RIG-1-dependent *IFN* transcription [131]. Moreover, another potential mechanism that could give an elevated IFN response after treatment with DNA repair inhibitors and high-LET radiation is the leakage of small, radiation-induced DNA fragments from the nucleus into the cytosol [107]. Due to the high ionization density of high-LET irradiation, small DNA fragments are produced at clustered damage sites. We speculate that the inhibition of repair may potentially enhance and prolong this effect, resulting in cGAS/STING-dependent IFN production after leakage of these fragments into the cytosol.

The ability of proton or carbon ion radiation to induce immunogenic cell death, as measured by the hallmark factors calreticulin, HMGB1, and ATP, has been demonstrated in several model systems [129,133,134]. Although multiple studies show similar induction of immunogenic cell death after low- and high-LET irradiation, certain studies have found that high-LET carbon ion (50–70 keV/µm) irradiation increases the secretion of HMGB1 to a greater level than lower-LET irradiation (13 keV/µm or lower) [134,135]. However, repair inhibitors were not addressed in these studies. As ATR and PARP inhibitors can increase X-ray-induced immunogenic cell death, they likely also induce immunogenic cell death to at least the same extent when combined with protons or carbon ions. Moreover, several studies have demonstrated the enhanced expression of the immune checkpoint protein PD-L1 also after proton and carbon ion irradiation [136,137,138]. The increase in PD-L1 was shown to be dependent on ATR and CHK1 after carbon ion irradiation [137,138], and the inhibitors of these kinases were able to impair the PD-L1 upregulation. This gives another rationale for combining high-LET irradiation with ATR or CHK1 inhibition.

Of note is that proton and carbon ion radiotherapy may induce less lymphopenia compared to classical radiotherapy with X-rays [139,140]. Lymphopenia leads to immunosuppression and is considered a major limiting factor for combination treatments of radiotherapy and immunotherapy. Irradiation with particles induces less lymphopenia because of the reduced radiation dose to the surrounding volumes of normal tissue. This will result in the sparing of circulating lymphocytes, lymph nodes, and/or bone marrow [141]. The reduced lymphopenia with carbon ions or protons will likely also be highly important upon co-treatment with DNA repair inhibitors, making it possible to achieve higher antitumor immune responses than for DNA repair inhibitors in combination with X-rays.

## 5. Conclusions and Future Perspectives

Recent studies have shown that low- and high-LET irradiation may cause different responses in the cells. The increased clustered DNA damage after high-LET irradiation may induce a different spectrum of DNA damage signaling events. Furthermore, the use of DNA repair pathways and the induction of antitumor immune signaling may be different for low- and high-LET irradiation. Based on the current knowledge, DNA resection and activation of the HR repair pathway seem to be increased upon high-LET compared to low-LET irradiation. As normal cells are more often non-cycling than cancer cells, targeting the repair pathways of cycling cells (e.g., HR repair) may partly provide tumor-specific radiosensitization. However, some of the preclinical studies conducted so far do not find that HR deficiency is associated with increased sensitivity to high-LET irradiation, and there are also conflicting results regarding the involvement of the NHEJ repair pathway. Of note is that HR repair occurs mainly in the S and G2 phases, and differences in cell cycle progression between the various cancer cell lines may thus be important. High-LET irradiation typically causes a stronger G2 checkpoint arrest compared to low-LET irradiation, and this could likely lead to the detection of more resection and HR repair, but the extent of arrest could vary between cell lines. It is also possible that high statistical uncertainty in some of the experimental assays (e.g., the clonogenic survival assay), or in the radiation dose measurements or LET values, may explain some of the discrepancies.

Better knowledge about the differences between responses at low- and high-LET could potentially provide a rationale for trials with so-called “LET painting”, a proton beam delivery technique to achieve a higher ionization density not only towards the distal end of the irradiation field but also in the center of the tumor [142]. Additionally, a better understanding of the relationship between cancer-associated genetic alterations and responses to low- and high-LET particle irradiation may help to guide patient selection for particle radiotherapy in the future.

It will be highly important to further investigate combination treatments of particle therapy with DNA repair inhibitors or other radiosensitizing drugs. Even for inhibitors showing similar sensitization of the tumor cells to low- and high-LET radiation, the tumor-specific radiosensitizing effects will likely be improved for particle radiotherapy. Due to the reduced radiation dose to the surrounding healthy tissue, there will be less radiosensitization of normal cells. High-throughput screening, directly comparing the effects of drugs in combination with X-rays versus protons or carbon ions, will likely be useful to identify the most efficient combinations. Notably, a joint activity of several repair pathways may be required to survive clustered DNA damage. Hence, a stronger and more specific sensitization to high-LET particle radiation may be obtained through drug combinations. Emerging preclinical work with traditional X-ray-based radiotherapy shows that DNA repair inhibitors, e.g., inhibitors of ATR or PARP, do not only lead to cellular radiosensitization, but can also enhance antitumor immune signaling in many cases. However, very little is known about antitumor immune signaling upon combination treatment with such inhibitors and carbon ion or proton radiation, and this should be explored in future studies evaluating both immunogenic and potential immunosuppressive effects. Finally, to promote clinical translation, further preclinical assessment of the combination treatments will be needed in immunocompetent mouse models. The evaluation of triple combinations of particle radiotherapy, radiosensitizing drugs, and immune checkpoint inhibition will also be valuable.

## Figures and Tables

**Figure 1 cells-13-01058-f001:**
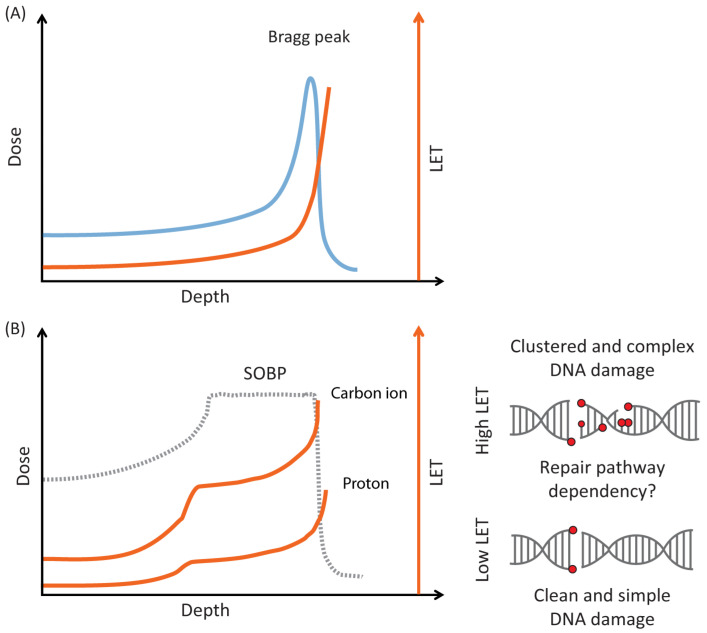
Illustration of depth dose and LET distribution of particle beams. (**A**) Pristine Bragg peak (blue curve) with increasing linear energy transfer (LET; orange curve) at the distal dose fall-off. (**B**) Spread-out Bragg peak (SOBP; dotted curve) with typical LET profiles for clinical carbon ion and proton beams (orange curves). Examples of DNA damage caused by high- and low-LET irradiation are shown to the right.

**Figure 2 cells-13-01058-f002:**
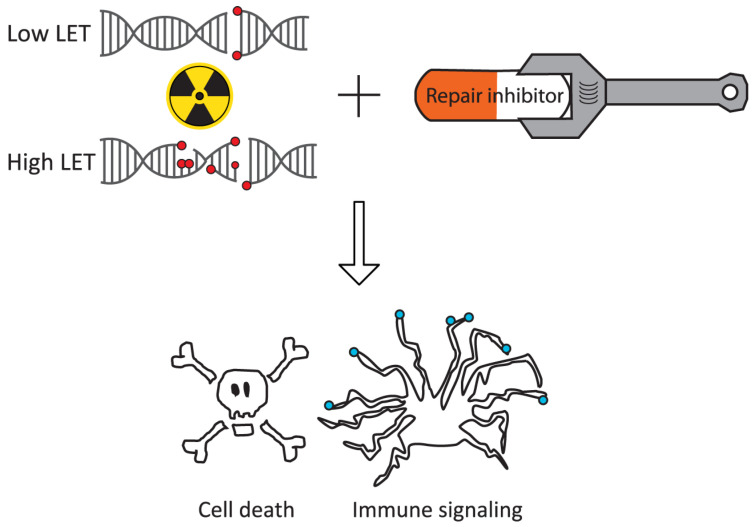
DNA repair pathway choice may be different for damage induced by low- versus high-LET irradiation. Consequently, the effects of DNA repair inhibitors on radiation-induced cell death and antitumor immune signaling may be different.

**Figure 3 cells-13-01058-f003:**
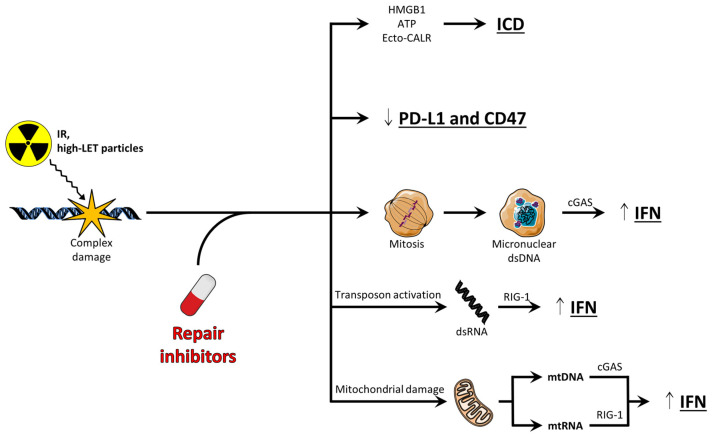
Potential mechanisms of increased antitumor immune signaling following treatment with DNA repair inhibitors and high-LET radiation. DNA repair inhibitors include inhibitors of cell cycle checkpoints. IFN: interferon; ICD: immunogenic cell death. Small, vertical errors indicate changes in occurrence of proteins.

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
