# Peer review of "Potential Benefits of Combining Proton or Carbon Ion Therapy with DNA Damage Repair Inhibitors"

_cells, 2024, doi:10.3390/cells13121058_

Round 1

Reviewer 1 Report

Comments and Suggestions for Authors

Potential benefits of combining proton or carbon ion therapy with DNA damage repair inhibitors  by Gro Elise Rødland and colleagues discuss potential differences in repair signaling between high- and low-LET irradiation and examine previous preclinical work involving DNA repair inhibitors combined with proton or carbon ion irradiation, proposing that DNA repair inhibitors are useful in combination with particle radiotherapy, causing tumor radiosensitization and enhancing antitumor immune effects. This reviewer suggests though some additional work/modifications in order to make the review worth of publishing:

1)     The review reads easily and has a good flow of information, although some points might be touched a bit superficially. Due to the lack of scientific evidence, some points are left a bit vague and might benefit from some additional speculations;

2)     A more detailed explanation of Bragg peak and SOBP is needed in the introduction, as the literature varies in this regard;

3)     Figure 1 is not very informative, needs more details about the pristine BP, also it needs detail about what happens to LET after the radiation passes the BP distal end;

4)     Figure 2 is not informative, and does not add any additional details. Either improve it or delete it;

5)     Line 184 to 190 needs references;

6)     Line 274 needs spell check;

7)     Line 311 introduces hypoxic condition as variable but it needs a bit more details about the effect of hypoxic condition on variable LET, if any;

8)     Line 332 needs clarification on whether is SOBP or pristine BP.

Author Response

Reviewer 1,

Comments and Suggestions for Authors

Potential benefits of combining proton or carbon ion therapy with DNA damage repair inhibitors  by Gro Elise Rødland and colleagues discuss potential differences in repair signaling between high- and low-LET irradiation and examine previous preclinical work involving DNA repair inhibitors combined with proton or carbon ion irradiation, proposing that DNA repair inhibitors are useful in combination with particle radiotherapy, causing tumor radiosensitization and enhancing antitumor immune effects. This reviewer suggests though some additional work/modifications in order to make the review worth of publishing.

Author reply:

                We appreciate that the reviewer find the text understandable and easy to read, and we are thankful for the suggestions for improvement. Below you can find our point-to-point answers to the comments.

1)     The review reads easily and has a good flow of information, although some points might be touched a bit superficially. Due to the lack of scientific evidence, some points are left a bit vague and might benefit from some additional speculations;

Author reply:

                We have revised the text with this comment in mind, and have particularly made revisions in section 4, “Anti-tumor immune signaling induced by DNA repair inhibitors ,“ where we have included a more elaborate discussion of some of the referenced work.

2)     A more detailed explanation of Bragg peak and SOBP is needed in the introduction, as the literature varies in this regard;

Author reply:

                We agree, and the introduction has now been updated according to the request, and further strengthened by an updated figure 1 (see comment 3).

3)     Figure 1 is not very informative, needs more details about the pristine BP, also it needs detail about what happens to LET after the radiation passes the BP distal end;

Author reply:

                Figure 1, with figure legend, has been updated, and has been separated into two panels, where panel A is new and shows the pristine BP. We hope that this made the figure more comprehensive. Though there are interesting differences between the LET profiles of protons and carbon ions beyond the distal end of the BP, we believe that adding this information may complicate our message. The figure is meant to be an illustrative supplement to the text, and show the following: 1) LET is lower at the beam entrance than at the end of the Bragg peak (both pristine and SOBP). 2) LET is higher for the more heavily charged carbon ions than for protons. 3) Particles with high LET induce more clustered DNA damage than low LET particles. 

4)     Figure 2 is not informative, and does not add any additional details. Either improve it or delete it;

Author reply:

                We had hoped that the figure illustrated the main questions addressed in this review. However, we understand that it may have been a bit unclear. We have made a new version of the figure and updated the figure legend, and believe that the message is now more clearly conveyed.

5)     Line 184 to 190 needs references;

Author reply:

                The text is now updated with additional references.

6)     Line 274 needs spell check;

Author reply:

                Mistake has been corrected.

7)     Line 311 introduces hypoxic condition as variable but it needs a bit more details about the effect of hypoxic condition on variable LET, if any;

Author reply:

                We agree, and specifics about oxygen levels have been included. We have also given a more detailed discussion about oxygen enhancement ratios.

8)     Line 332 needs clarification on whether is SOBP or pristine BP.

Author reply:

                We agree that more specific experimental detail should be listed for this reference due to the significance of its findings. The radiation set-up has now been more thoroughly explained. 

Reviewer 2 Report

Comments and Suggestions for Authors

Nice and comprehensive review of the field.

I have just a couple comments / suggestions:

     Introduction

-          Why is there such a massive difference in LET depending on the position within SOBP for protons (~ 7x?), while “minimal” for carbon?  I would expect it to get higher, e.g. https://doi.org/10.1016/j.bbrc.2024.149500 (the SOBP generation is similar, no? ions of various energies)

-          Figure 2 – very generic – necessary?; does not clearly show what is written in its caption – maybe update / edit it?

Comments on the Quality of English Language

Minor details

-          183 – “to” not necessary (maybe reword the sentence?)

-          447 – end of the sentence – I don’t understand the meaning, possibly worth to rewrite the sentence?

Author Response

Reviewer 2,

Comments and Suggestions for Authors

Nice and comprehensive review of the field.

Author reply:

                We are very happy to hear that the reviewer finds our paper comprehensive, and we are grateful for the suggestions for improvement; a point-to-point response can be found below.  

I have just a couple comments / suggestions:

     Introduction

-          Why is there such a massive difference in LET depending on the position within SOBP for protons (~ 7x?), while “minimal” for carbon?  I would expect it to get higher, e.g. https://doi.org/10.1016/j.bbrc.2024.149500 (the SOBP generation is similar, no? ions of various energies)

Author reply:

                We agree, and have updated the text for carbon ions accordingly: “…with typical LET values within the spread-out Bragg peak (SOBP) of 40-80 keV/µm [4] and >200 keV/µm at the distal end , depending on the beam energy”. We have also cited relevant work including the suggested reference above.                 

-          Figure 2 – very generic – necessary?; does not clearly show what is written in its caption – maybe update / edit it?

Author reply:

                We have made an updated version of the figure and modified the caption to make them correspond better. See also reply to comment 4 of reviewer 1.

Comments on the Quality of English Language

Minor details

-          183 – “to” not necessary (maybe reword the sentence?)

Author reply:

                The sentence has been reworded “…sensitizes cells to PARP inhibitors…”

-          447 – end of the sentence – I don’t understand the meaning, possibly worth to rewrite the sentence?

Author reply:

                We have rephrased this sentence, and added more explanation to the previous sentence.